# Design of Kinematic Connectors for Microstructured Materials Produced by Additive Manufacturing

**DOI:** 10.3390/polym13091500

**Published:** 2021-05-06

**Authors:** Miguel R. Silva, João A. Dias-de-Oliveira, António M. Pereira, Nuno M. Alves, Álvaro M. Sampaio, António J. Pontes

**Affiliations:** 1CDRSP, ESTG, Polytechnic of Leiria, 2401-951 Leiria, Portugal; mario.pereira@ipleiria.pt (A.M.P.); nuno.alves@ipleiria.pt (N.M.A.); 2Institute for Polymers and Composites—IPC, School of Engineering, University of Minho, 4800-058 Guimarães, Portugal; amsampaio@dep.uminho.pt (Á.M.S.); pontes@dep.uminho.pt (A.J.P.); 3Department of Mechanical Engineering, TEMA/Centre for Mechanical Technology and Automation, University of Aveiro, 3810-193 Aveiro, Portugal; jalex@ua.pt; 4Lab2PT, School of Architecture, University of Minho, 4800-058 Guimarães, Portugal

**Keywords:** kinematic connectors, functionally graded materials, microstructured, additive manufacturing, topology optimisation

## Abstract

The main characteristic of materials with a functional gradient is the progressive composition or the structure variation across its geometry. This results in the properties variation in one or more specific directions, according to the functional application requirements. Cellular structure flexibility in tailoring properties is employed frequently to design functionally-graded materials. Topology optimisation methods are powerful tools to functionally graded materials design with cellular structure geometry, although continuity between adjacent unit-cells in gradient directions remains a restriction. It is mandatory to attain a manufacturable part to guarantee the connectedness between adjoining microstructures, namely by ensuring that the solid regions on the microstructure’s borders i.e., kinematic connectors) match the neighboring cells that share the same boundary. This study assesses the kinematic connectors generated by imposing local density restrictions in the initial design domain (i.e., nucleation) between topologically optimised representative unit-cells. Several kinematic connector examples are presented for two representatives unit-cells topology optimised for maximum bulk and shear moduli with different volume fractions restrictions and graduated Young’s modulus. Experimental mechanical tests (compression) were performed, and comparison studies were carried out between experimental and numerical Young’s modulus. The results for the single maximum bulk for the mean values for experimental compressive Young’s modulus ($E^{\bar{x}}$) with 60$\%\;V_{f}$ show a deviation of 9.15%. The single maximum shear for the experimental compressive Young’s modulus mean values ($E^{\bar{x}}$) with 60$\%\;V_{f}$, exhibit a deviation of 11.73%. For graded structures, the experimental mean values of compressive Young’s moduli ($E^{\bar{x}}$), compared with predicted total Young’s moduli ($E^{\mathrm{Se}}$), show a deviation of 6.96 for the bulk graded structure. The main results show that the single type representative unit-cell experimental Young’s modulus with higher volume fraction presents a minor deviation compared with homogenized data. Both (i.e., bulk and shear moduli) graded microstructures show continuity between adjacent cells. The proposed method proved to be suitable for generating kinematic connections for the design of shear and bulk graduated microstructured materials.

## 1. Introduction

The study and design of Microstructured Periodic Materials (MPMs) with specific properties is currently one of the most promising areas in the research of new materials. Natural materials have served as an inspiration/basis for the design of new artificial materials (i.e., designed and produced by man, not available in nature) with high performance [1,2] and multifunctional responses [1,3,4]. MPMs are an area of special interest, as they allow great versatility in the design of materials with specific properties, by controlling the composition of the phases (i.e., materials) and the internal microstructure. MPMs are formed by a number of identical Representative Unit Cells (RUCs), where the internal distribution of the phases determines their properties [5]. MPMs with gradient properties (i.e., Functionally Graded Materials (FGMs)) are a class of materials whose physical properties gradually vary in one or more directions [6]. Currently, there is a tendency to use the Topology Optimisation (TO) method, in the design of MPMs [6,7,8]. In addition, the accelerated development of Additive Manufacturing (AM) technology has overcome the limitations of conventional manufacturing processes. The AM has emerged as a valid technology for the production of MPMs (or structures), with great geometric freedom [9,10], complex materials and functionality, e.g., FGMs [11]. However, a problem that results from this approach is the lack of connectivity between adjacent cells that may prevent the materialization of MPMs. This problem can be addressed using kinematic connectors, which by definition are solid elements placed in the matching boundary location between two adjacent RUCs (Figure 1). Ensuring the kinematic link between adjacent RUCs in the design phase and their experimental evaluation is fundamental topics, but rarely addressed in the literature and remains an open topic.

In recent years, MPMs have gained prominence in several areas of engineering [12]. These porous materials are characterized by high performance with low weight. MPMs, are made of RUCs or Representative Volume Elements (RVEs), in which the manipulation of the constituent phases distribution, more than the composition of the materials that compose them, allows for adjusting the properties of the materials and their response to external requests [5,6,13]. Therefore, the distribution of phases in the admissible design space offers the chance to conceive Multifunctional Microstructured Periodic Materials (MPMMs), through advanced design techniques such as TO [14,15].

To date, several methods for TO have been developed, namely the homogenization method [16,17,18], the Evolutionary Structural Optimisation (ESO) method [19,20] or Bidirectional ESO (BESO) [21,22], Genetic Algorithm (GA) [23,24,25], and the Level Set Methods (LSM) [26,27,28]. Another popular method is the Solid Isotropic Material with Penalization (SIMP); this numerical FEM based uses a continuous density approach with fixed mesh (Eulerian), in order to solve the TO problem—typically by discretizing the design domain into a large number of finite elements (*N*) and the density distribution being expressed by *N* element design variables [29,30,31,32]. Another approach is the AM shape optimisation method that takes into account AM restrictions in an early part design stage. Recently, Lianos et al. [33] propose a novel optimisation method that simultaneously takes into account the structural stress and AM constraints in the initial part geometry. In this method, the AM buildability constraints are addressed in parallel with the material optimisation.

TO was initially developed and applied in the conception and design of structures. However, if considering MPMs formed by two scales, the macroscale formed by the global structure and the microscale formed by the RUCs can extend the application of TO to the individual optimisation of each of the RUCs, applying appropriate periodic boundary conditions [6]. Usually, TO is performed in conjunction with homogenization methods for approximate calculation of RUC properties. However, the TO formulation combined with MEF is susceptible to suffering from several problems related to numerical instabilities, such as checkerboard, mesh dependency, and local minimums [34].

Another associated problem is the constant fields caused by the imposition of deformation states with a single material phase, which give rise to localized fields and constant sensitivities in the entire finite element mesh and, consequently, the variables updating is done in block. Although it is not a common problem, it is something that arises in some cases, such as in problems of inverse homogenization. This issue can be overcome by imposing some kind of numerical instability or by imposing the initial fields (e.g., density) [35]. The imposition of elements of maximum density in the initial design, allows for destabilizing the problem and facilitating the updating of the variables.

The mechanical, chemical, and physical behavior of MPMs (i.e., effective or homogenized properties) depends on the topology of the internal microstructure, volume fraction, and properties of its constituents. Interactions between components must also be considered, such as failure in the interface of the components or damage resulting from fracture of the constituents. Thus, the precise characterization of the behavior of composite and cellular materials is a complex issue that requires adequate and sophisticated methods [36].

Several theories have been developed to predict the behavior of composite and cellular materials. A common approach to connect the two scales (i.e., micro and macro) and model the response of composite and microstructured materials is to use the average of the properties of their constituents. The simplest forecasting rules include the classic Voigt/Reuss limits, such as the rule of mixtures [37,38] and Hill [39]. These rules are often used for their simplicity, in predicting the effective elastic properties of multiphase composite materials. However, these rules are based on assumptions of uniform strain or stress fields and volume fractions. Therefore, the morphology, topology, and spatial distribution of the constituents in the volume of the material are not taken into account. These aspects can lead to significant uncertainty in determining the properties of composite and cellular materials [40]. Another approach is based on the mechanics of materials. Numerous researchers have used the formulation of material mechanics based on analytical methods or the Finite Element Method (FEM) to calculate the effective elastic properties of composite and cellular materials [12,41,42,43,44,45,46].

A common approach, homogenization, consists of replacing the detailed modeling of the heterogeneous material that makes up a given structure with an equivalent homogeneous means, according to the laws of microstructural behavior obtained from the analysis of constitutive information calculated according to the representative behavior of heterogeneous units of material [47]. Direct Computational (or Numerical) Homogenization, also referred to in the literature as Representative Volume Element (RVE) homogeneization method, is another way of estimating the properties of these complex materials. In this approach, the effective elastic properties of a microstructure are calculated directly through the discretization of the admissible design domain and using the FEM.

Several authors have presented studies in which they determine the RUC-based effective properties of periodic structures [47,48] and non-periodic [49].

In the last decade, AM has emerged as a viable technology to overcome the limitations of conventional manufacturing processes, in particular because of its ability to produce complex geometries, allowing great freedom in shapes, materials, and complex functionalities [11]. However, the real advantage of AM technology is when product performance is maximized through the synthesis of shapes, sizes, structural hierarchy, and composition [50]. For these reasons, AM is a valid technology, namely for the production of microstructured polymer composites (or structures) with optimized design, of greater complexity, with new specific properties and behaviors.

There are several polymer AM processes, namely Material Jetting (MJ), like the process PolyJet^TM^ proprietary of Stratasys, which uses thin piezoelectric nozzles to deposit drops of photosensitive resin (thermosetting polymer), which are cured by UV light [51,52]. The nature of the PolyJet^TM^ process makes it possible to produce heterogeneous materials composed of discrete voxels (i.e., discrete volumes of material).

FGMs are characterized by having physical properties that vary gradually, in one or more directions, depending on the functional requirements of the project [53]. In these non-homogeneous materials, the properties of the gradients are obtained by changing their composition or microstructure in their volume [54]. In nature, one can find several examples of FGMs such as bamboo, shell, tooth, and bone [55]. In these materials, biological structures change their layer-by-layer mechanical properties by varying their constituents and the internal microstructure, as a way of adapting (i.e., responding) to external stresses [56,57]. In this way, FGMs can adapt their response to requests and boundary conditions defined by service requirements in order to enhance their mechanical performance e.g., stiffness and strength [54]. FGMs allow the design of new materials with gradients of properties and multifunctionality, for application in various industries such as automotive, health, aeronautics, and aerospace [58,59,60,61]. An additional relevant theme is the actual AM parts’ mechanical performance. The AM printed parts exhibit anisotropic mechanical properties due to the several aspects related to the process nature and to production parameters (e.g., build orientation, deposition speed). The anisotropic mechanical properties of the printed AM parts affect the final component performance. To increase these parts’ industrial adoption, it is necessary to characterize the parts’ mechanical behavior (i.e., tensile, flexural, compression, impact, and shear) to determine actual service performance. Several Standards can be followed for conducting those tests on polymers-based parts [62].

Another main topic is the trade-off between TO and manufacturability based on necessary resources (i.e., energy, cost, time) to produce the final product. This can be addressed during the initial steps before the TO procedure. In a recent study, Lianos et al. [63] propose an assessment method to screen the added value that an AM technology can offer to an entity by quantifying the AM utilization throughout the distinct product development and production stages.

However, most studies carried out on the design of microstructured FGMs focus on the individual design of RUCs with different volume fractions. One of the problems resulting from this approach is the connection between adjacent RUCs, especially in the direction of the gradient (i.e., request). It is necessary to ensure that the solid regions on the borders of the RUCs are coincident with the RUCs that share the same border, for the kinematic transmission to occur. Despite being a topic rarely addressed in the literature, Zhou and Li [64,65] proposed three approaches to ensure connectivity between adjacent RUCs of FGMs; (i) non-design, (ii) pseudo loading, and (iii) unified formulation with nonlinear diffusion. The connective constrain method assigns some non-designable voxels on the external faces (boundaries) of each RUC. However, this approach forces the optimal solution dependency on the non-designable voxels selection. The second proposed method, pseudo load, defines a pseudo load and a kinematic boundary condition prescribed along the gradient direction. However, the weighting factor of the pseudo load should be selected carefully to avoid a dominant role in the optimal design. Lastly, for the unified formulation with a nonlinear diffusion method, Zhou and Li [64] applied the nonlinear diffusion technique [66] as a filtering scheme in combination with the SIMP method.

Radman et al. [7] proposed a new Optimisation algorithm based on the BESO method for design FGMs. The authors reported proper connections between adjacent RUCs obtained with low computational cost. Instead of designing a series of base microstructures simultaneously, the base cells are optimized sequentially, considering three base cells at each stage.

The authors overcame this problem by applying local density restrictions, in predetermined elements of the boundary of the initial admissible design domain (i.e., initial RUC). This restriction can be implemented by assigning density 1 (rigid material) to four elements on each face of the initial RUC (Figure 2). This density constraint allows, on the one hand, to create instability in the OT algorithm and facilitate the start of the process of updating the densities of the elements, and on the other hand to ensure that the initial points of density 1 in the iteration *i* = 0 serve as nucleation points, around which gradients form, leading the material to grow and ensuring kinematic connections between adjacent RUCs.

This work aims to present a design method used in the TO procedure to guarantee linkage between adjacent CRUs.This design methodology could solve an important issue regarding the RUC combination viability design, something that is very seldomly approached. More can contribute to the effective adoption of periodic microstructure cellular materials for designing new materials with specific properties (e.g., lightweight, functional gradients, metamaterials). This study proposes a method to generate kinematic connectors by imposing local density restrictions in the TO RUC initial design domain (i.e., nucleation). The method herein can be used for industrial engineering applications in several areas (e.g., health, automotive industry, aeronautics). This study assesses the kinematic connectors, generated by imposing local density restrictions in the initial design domain (i.e., nucleation), between topologically optimized RUCs. Different examples of Kinematic Connectors (KCs) are presented: (a) RUCs with the same mechanical properties and (b) RUCs with identical mechanical properties (same family) with a properties gradient. The property gradient studied, Young’s modulus, is achieved by varying the volume fraction restriction in the TO. A comparative study is also presented, between the homogenized Young’s modulus obtained through DCH and experimentally of graduated polymeric microstructures produced by PolyJet™.

## 2. Materials and Methods

Finite element software Abaqus CAE (Dassault Systèmes, Simulia Corp, Providence, RI, USA) was used to model the cell. This choice was because the TO procedure implemented in a Python script was dependent on an external finite element software. An initial design domain (cubic RUC) is discretized in 20 × 20 × 20 (8000 elements) and 9261 nodes using eight-node brick elements (Abaqus C3D8R, linear brick element with reduced integration) (Figure 2). The CRU discretization (i.e., number of elements) and shape brick element type were selected to reduce computational costs. In order to avoid singularities in FEM problem, when solving equilibrium equations in an entire design space, a minimum density lower bound is imposed ($0<\rho_{\min}\leq \rho$) through the parametrization of two materials (i.e., rigid and soft). The properties, Young’s modulus, and Poisson’s ratio, of the two phases (i.e., materials) constituting the RUC were defined in Table 1. The Young’s modulus (*E*${}^{1}$) and Poisson’s ratio (ρ) were obtained through experimental mechanical tests. Density 1 was imposed on four elements on each side of the RUC (Figure 2).

The RUC has been topologically optimized with hydrostatic and shear uniform displacements (Figure 3), with imposition of Periodic Boundary Conditions (PBCs) and volume restrictions of 0.1–0.6, using the SIMP algorithm with Optimal Criterion (OC) and the Objective Function (OF) set to minimum compliance [67]. The TO procedure was run with typically proven SIMP parameters, penalty, filter radius, and gray-scale filter (Table 1) to reach a convergence solution [34,68,69]. The optimisation was carried out through a Python script and executed in the Abaqus scripting environment.

After the TO procedure, the number of solid elements of the RUC mesh was extracted from the .cae file (Abaqus) and recorded in .csv files, having subsequently been imported by a Python script to generate the .stl file of the RUC assembly. The RUCs were modeled with the unit dimensions 10 × 10 × 10 mm^3^ (Voxel 500 × 500 × 500 μm^3^) and assembled in a 2 × 2 × 6 arrangement of unit cells (Figure 4 and Figure 5). The graded structures with volume fractions from 0.1 to 0.6 and Young’s modulus of 70.53, 171.43, 280.76, 445.41, 705.33, 919.99 for the maximum bulk case and 25.55, 50.68, 112.56, 281.36, 431.08, 605.14 for maximum shear were assembled in the same arrangement 2 × 2 × 6 unit cells (Figure 6).

### 2.1. Direct Computational Homogenization

The concept of DCH is based on the numerical imposition of a field of uniform displacements to calculate the elastic properties of composite materials. These displacements are imposed in several independent sets, one for each elastic property of the material (E11, E22, E33, G12, G23 and G13), for a total of six for a 3D RUC. It is assumed that the RUC is part of a periodic material, so it is necessary to simulate the periodicity of the RUC with material from its surroundings. Initial studies reached the periodicity through the imposition of boundary conditions that guaranteed that the plane of the RUC surfaces remained flat after the deformation [70,71]. In fact, this is only valid for cases in which normal loading is applied at RUC borders. In the case of shear loading or orthotropic representation, they are not valid, since the excessive restriction of degrees of freedom (over-constraining) of the RUC leads to overestimating the elastic properties of the composite [72,73,74,75]. Thus, it is necessary to apply periodic boundary conditions so that the surfaces of the RUC boundaries can deform (i.e., without remaining flat) [73,76]. To impose these periodic conditions, it is necessary to connect the degrees of freedom of the nodes of the opposite surfaces [75,77]. DCH was implemented through a Python script within the Abaqus Scripting environment [78,79].

The effect of nucleations on the amount of solid elements generated at the RUC’s borders (i.e., faces) was evaluated. For this purpose, a metric named Solid Elements per Face Ratio (SEFR) was established, which calculates the ratio between the number of solid elements of each face by the total elements of the face.

### 2.2. AM and Mechanical Characterization

A samples test was produced in VeroClear RGD810™ thermosetting (Stratasys Inc., Minneapolis, MN, USA) polymer (rigid phase) and the base supports in SUP706B in a Objet30 Prime (Stratasys Inc., Minneapolis, MN, USA), with a resolution of 32 μm per layer in a single batch. Process parameters were kept constant during the build process. The materials were supplied by the company Stratasys™. Production was carried out under the following conditions: (i) automatic positioning; (ii) “gloss mode” option (i.e., glossy, without supporting material to wrap the piece); (iii) the resins were stored in a controlled environment, to be placed previously in the equipment, according to the supplier’s rules; (iv) supports were removed in a chemical bath of 2% Sodium Hydroxide (NaOH) and 1% Sodium Metasilicate (Na_2_SIO_3_); (v) mechanical tests were performed on samples as produced (no further oven curing).

#### Mechanical Tests

Uniaxial quasi-static compression mechanical tests were performed using an Instron model 4505 universal machine (Instron Worldwide, Norwood, MA, USA) according to the ISO 844 standard [80] at an ambient temperature of 23 °C and a crossbar feed rate of 1 mm/min. The force was measured with a 5 and 100 kN load cell of the testing machine, and the displacement of the machine plate was used to determine the axial strain in the specimen. Furthermore, samples were prepared for each base CRU with three volume fraction 10, 30, 60$\%\;V_{f}$ and for the two graded samples 10–60$\%\;V_{f}$ with maximum bulk and shear case, respectively. Three test samples were tested for each base RUCs EV10, EV30, EV60, GV10, GV30, GV60, and for each graded sample GE1060, GG1060, for a total of 24. The Young’s modulus of each specimen was calculated considering the stress values at the extension points 0.0005 and 0.0025 mm/mm. The primary force-displacement data obtained from the recording software were normalized to stress–strain data.

For the calculation of theoretical total Young’s modulus of the graded structures (*E*${}^{\mathrm{Se}}$), consider that the series arrangement of *n* springs is given by Equation (Equation 1),
(1)$$\frac{1}{K_{\mathrm{tot}}}=\sum_{i=1}^{n}\frac{1}{k_{i}}$$
where *K*${}_{\mathrm{tot}}$ is the total stiffness, and *k*${}_{i}$ the stiffness of each spring. Applying this formula to a graded structure (Equation (Equation 2)) composed of *n* materials with Young’s modulus *E*${}_{i}$, stacked in the same direction of the applied load, can be considered an approximation for the total Young’s modulus of the graded structure (assuming the cross section area is constant throughout the loading direction):(2)$$\frac{1}{E^{\mathrm{Se}}}=\frac{1}{n}\sum_{i=1}^{n}\frac{1}{E_{i}}$$

## 3. Results and Discussion

The experimental work was all done using results from the topology optimisation, with special care to approximate mesh and printing resolutions. Furthermore, the RUCs’ volume fraction resulted from TO volume restrictions. The final kinematic connectors shape results from both the initial location’s density restrictions and TO procedure.

Compression mechanical tests were conducted to compare experimental Young’s modulus with the theoretical obtained by DCH. The $E^{E}$ value has been determined, considering the first linear portion of the stress–strain curve and estimating its slope by linear fitting procedures (Figure A3). The Young’s modulus values have been estimated as averages over three replications of each of the eight types (Figure 4, Figure 5 and Figure 6).

Comparisons of Young’s modulus between experimental and homogenized for hydrostatic and shear cases single cells and for the two graded structures are shown in Table 2.

### 3.1. Single Maximum Bulk RUCs

The number of iterations necessary to reach the OF convergence (i.e., solution) is dependent on both the parameters used in the TO algorithm and the established convergence criterion. For the TO step, the total number of iterations varied between 59 and 40 for 10% and 20$\%\;V_{f}$, respectively (Table 3 and Figure A4a).

The face elements of RUCs with density restrictions (ρ=1) acted as initial nucleations and triggered the growth of solid elements around them. Figure 7 illustrates the evolution of the solid element regions around the initial nucleations according to the increase in volume fraction. At intervals 10–30$\%\;V_{f}$, the growth of the solid regions occurs only around the initial nucleations, between 40–60$\%\;V_{f}$, the regions of solid elements merge together, this fact coincides with the increase of the RUC’s experimental compressive Young’s modulus ($E^{\bar{x}}$) and bulk modulus (*K*). The SEFR ratio assesses the number of solid elements at the RUC border, and this parameter is critical to assure that kinematic connectivity between adjacent RUCs higher SEFR ratio will be desirable (i.e., a very rigid element will be zero stiffness in a global assembly if there is no solid element connection to the neighbors). For the single maximum bulk case, the increase in $V_{f}$ also corresponds to a monotonic increase in solid elements on the face of the RUCs (Table 3 and Figure 8a).

The homogenization was carried out for each one of the six different 10–60$\%\;V_{f}$ base cells. The homogenized Young’s modulus of bulk RUCs ($E^{H}$) varies from 70.53 to 919.99 MPa from 10 to 60$\%\;V_{f}$, respectively. The experimental mean compressive Young’s modulus ($E^{\bar{x}}$) showed values of 47.09, 251.95, and 1012.69 MPa for the 10, 30, and 60$\%\;V_{f}$, respectively (Table 2). The bulk modulus (*K*) shows a significant increase for $V_{f}$ greater than 40%; moreover, it assumes an almost linear behavior between 40–60$\%\;V_{f}$ (Table 3 and Figure 8a). This behavior, as mentioned before, may be due to the union of the four nucleation regions (i.e., rigid material) at the border of the RUC (i.e., faces) (Figure 7). The standard deviation (*S*) varies between 3.29 and 28.32 for 30 and 60$\%\;V_{f}$, respectively (Table 2).

Figure 9 shows the experimental mean stress–strain diagram for the maximum bulk case for the three volume fractions. These curves have characteristics of a semi-rigid polymer with an elastic region, however, without showing a well-defined transition to plastic deformation region, typically observed on VeroClear Polyjet material [51,81,82].

The mean values for experimental compressive Young’s modulus ($E^{\bar{x}}$) with 10, 30 and 60$\%\;V_{f}$, when compared with homogenized values, show a deviation of −49.77, −11.43 and 9.15%, respectively (Table 2). First, the deviation decreases for higher $V_{f}$ cells; these negative deviations for the lower volume fractions may be related to the plastic instability observed during the compression tests. Figure 10b shows a buckling effect occurred in 10$\%\;V_{f}$ sample test that may be responsible for the high Young’s modulus relative error percentage. To prevent this, the sample test thickness should increase to reduce the aspect ratio (i.e., $\frac{height}{width}$); additionally, a thin layer of solid material in the sample test contact area with the machine’s plates can be printed. Second, the experimental 60$\%\;V_{f}$ sample exceeded the homogenized value. This may due the fact that the 60$\%\;V_{f}$ unit cells were tested with the support material inside, due to the absence of holes that allows its removal during post-processing. For more details, Figure A1 shows the topological optimized RVEs for maximum bulk modulus and corresponding homogenized elasticity matrices.

### 3.2. Single Maximum Shear RUCs

For this case, the number of iterations required to obtain the final topologies varied between 40 and 51 for 10% and 60$\%\;V_{f}$, respectively (Table 3 and Figure A4b). Similar to the previous case (Bulk), the initial inclusions served with nucleus elements. However, in this case, the regions of the solid elements are united for all the volume fractions. Furthermore, it is in the central region of the RUC that there is a more pronounced evolution in the number of solid elements with the increase in $V_{f}$ (Figure 11). This may indicate that the proposed method, despite restricting the design domain, does not overlap excessively (i.e., does not dominate over TO) in the distribution of rigid elements in areas with higher sensitivities. The SEFR ratio presents higher values for all volume fractions compared to the bulk case. However, this difference decreases with the increase in $V_{f}$. Furthermore, the evolution of SEFR to 40$\%\;V_{f}$ has a negative value, compared to the previous value of 30$\%\;V_{f}$ (Table 3 and Figure 8b).

The homogenization was carried out for each one of the six different 10–60$\%\;V_{f}$ RUCs. The homogenized Young’s modulus of shear RUCs ($E^{H}$) vary from 25.55 to 605.14 MPa from 10 to 60$\%\;V_{f}$, respectively (Figure 8b). Table 2 shows the mean experimental compressive Young’s moduli ($E^{\bar{x}}$) with values of 10.54, 83.11, and 685.60 MPa for 10, 30, and 60$\%\;V_{f}$, respectively. The shear modulus (*G*) shows a significant increase for $V_{f}$ greater than 30%, moreover, it assumes an almost linear behavior between 30–60$\%\;V_{f}$
Table 3. This behavior coincides, in a first phase (30$\%\;V_{f}$), with the increase in the number of solid elements in the center of the face of the RUC and in the second phase (50$\%\;V_{f}$) with an increase in elements in the crossed arms that join the nucleations initials to the center of the RUC. The experimental compressive Young’s modulus standard deviation (*S*) varies between 0.85 and 3.66 for 10 and 60$\%\;V_{f}$, respectively (Table 2).

Figure 12 shows the experimental mean stress–strain diagram for the maximum bulk case for the three $V_{f}$. These curves have characteristics of a semi-rigid polymer with an elastic region, however, without showing a well-defined transition to plastic deformation region, typically observed on VeroClear Polyjet material [51,81,82].

For the shear case, the mean values for experimental compressive Young’s modulus ($E^{\bar{x}}$) with 10, 30, and 60$\%\;V_{f}$, when compared with homogenized values, show a deviation of −142.41, −35.43, and 11.73%, respectively (Table 2). First, the deviation decreases for higher $V_{f}$ cells. These negative deviations for the lower % volume fractions may be related to the plastic instability observed during the compression tests. Figure 10c shows a shear sample test with 10% $V_{f}$ after a compression test. The failure occurred in contact RUCs with the test plate. This effect may be caused by friction with plates that generate a not deformable material cone. That may be responsible for the high Young’s modulus relative error percentage. To prevent this, a thin layer of solid material can be printed in the sample test contact area with the machine’s plates and should increase the sample test thickness. Second, the experimental 60$\%\;V_{f}$ sample exceeded the homogenized value. This may due the fact that the 60$\%\;V_{f}$ unit cells were tested with the support material inside, due to the absence of holes that allow its removal during post-processing. Compared with the hydrostatic case, the shear single cells show similar behavior, with a much higher error. For more details, Figure A2 shows the topological optimized RVEs for maximum shear modulus and corresponding homogenized elasticity matrices.

### 3.3. Graded Structures

The predicted total Young’s modulus ($E^{\mathrm{Se}}$) of the bulk and shear graded structure showed values of 141.24 and 53.13 MPa, respectively. The experimental mean values of compressive Young’s modulus ($E^{\bar{x}}$) of graded structures reached values of 151.80 and 37.70 MPa with standard deviation (*S*) of 10.88 and 3.57 MPa, for the bulk and shear case, respectively. The experimental mean values of compressive Young’s moduli ($E^{\bar{x}}$) of graded structures were compared with predicted total Young’s moduli ($E^{\mathrm{Se}}$) computed with Equation (Equation 2). For the bulk and shear graded structures, the errors were 6.96 and −40.90%, respectively. The failure modes for both graded structures occurred in the lowest volume fraction RUC and are similar to those observed for corresponding single RUCs. Figure 10f,h shows maximum bulk and shear grade structures, respectively after testing. The failure occurred due to buckling and not deformable cones for the bulk and shear case, respectively. Figure 13a–d show the graded sample tests solid model and the interface between unit cells with different volume fractions, for the maximum bulk and shear case, respectively. In Figure 13b,d, the connection zones between adjacent RUCs show that connectivity is maintained, which enables the kinematic connection through the entire graded structure.

Figure 14a,b show the experimental mean stress–strain diagram for the maximum bulk and shear case graded structures, respectively. In Figure 14a, the stress–strain curves show disparities in the maximum stress values; this may be due to the failure mode described previously (buckling) and its unpredictable nature.

## 4. Conclusions

This paper presented a method to ensure a kinematic connection in designing microstructured periodic materials with graded Young’s modulus. The connectivity is achieved by the imposition of local density restriction in the initial design domain. A SIMP based Topology Optimisation method subjected to a volume constraint had used for the computational design of RUC unit cells for maximum bulk and shear modulus. The RUCs homogenized Young’s moduli were computed through direct computational homogenization. Two graded structures were assembled with graduated Young’s modulus. The single RUCs with maximum bulk, shear, and graded structures in the 4×6 arrangement were produced by AM-PolyJet. Mechanical compression tests were performed, and an experimental Young’s modulus was calculated. A comparison between experimental and theoretical data was conducted.

For the single maximum RUCs, the mean values for experimental compressive Young’s modulus ($E^{\bar{x}}$) with 10, 30, and 60$\%\;V_{f}$, when compared with homogenized values, show a deviation of −49.77, −11.43 and 9.15%, respectively. In this group, the principal failure mode was buckling.

Considering the single maximum shear RUCs, the experimental compressive Young’s modulus mean values ($E^{\bar{x}}$) with 10, 30, and 60$\%\;V_{f}$, when compared with homogenized values, show a deviation of −142.41, −35.43 and 11.73%, respectively. In this group, the principal failure mode was caused by RUC friction with plates that generates a not deformable material cone. The results show that a single type RUC experimental Young’s modulus with a higher volume fraction presents a smaller deviation compared with homogenized data.

For graded structures, the experimental mean values of compressive Young’s moduli ($E^{\bar{x}}$), compared with predicted total Young’s moduli ($E^{\mathrm{Se}}$), show a deviation of 6.96 and −40.90% for the bulk and shear graded structures, respectively. The failure modes for both graded structures occurred in the lowest volume fraction RUC and are similar to those observed for corresponding single RUCs. Although the high volume fraction gradient for the graded microstructured shows continuity between adjacent cells. The proposed method proved to be suitable for generating kinematic connections for the design of shear and bulk graduated microstructured materials.

It should be pointed out that, although almost all sort of gradients could be achievable theoretically, the mesh size discretization and the printer resolution (i.e., voxel size) may restrict the experimental gradients. The RUCs presented herein have specific dimensions according to the PolyJet™process minimum printing feature. However, it can be discretized with different mesh sizes (i.e., changing the mesh elements number in the initial design domain) and scale for a specific applications. The presented method were validated with the Polyjet process with VeroClear (rigid polymer) but can be extended to other AM process and materials (e.g., Fused Filament Fabrication, Stereolithography, Selective Laser Sintering). Furthermore, the proposed method is a high-level approach that does not depend on specific AM technology or material.

The advantage of the proposed method is its ease of implementation and the potential to be extended to design new materials with specific properties (e.g., lightweight, functional gradients, metamaterials) for industrial engineering applications in various areas (e.g., health, industry automotive, aeronautics).

## Figures and Tables

**Figure 1 polymers-13-01500-f001:**
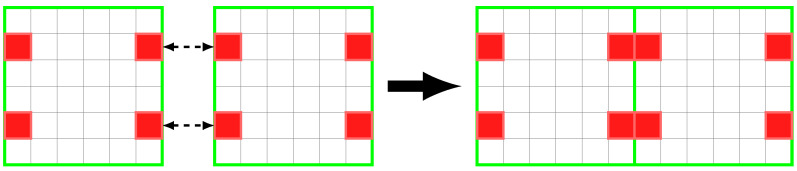
Two adjacent RUCs, highlighted in red are the kinematic connectors responsible for local kinematic transmission between contiguous RUCs.

**Figure 2 polymers-13-01500-f002:**
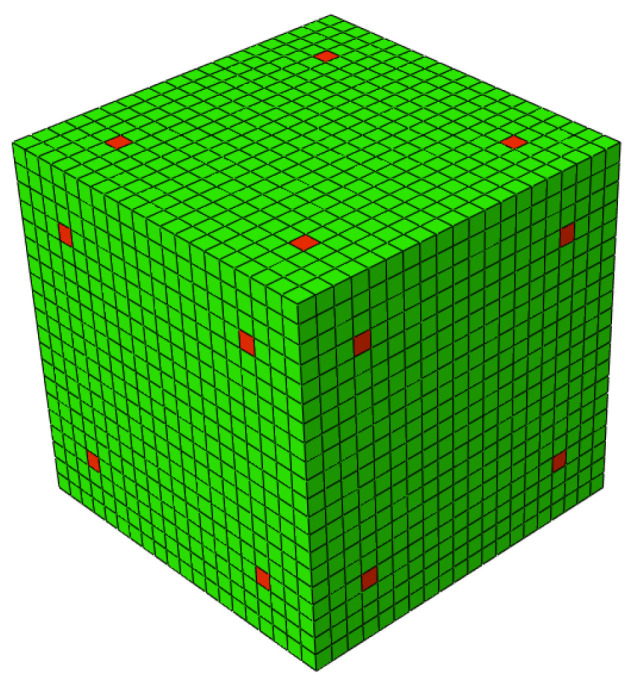
Initial design, the red elements on the faces of the RUC have ρ = 1.

**Figure 3 polymers-13-01500-f003:**
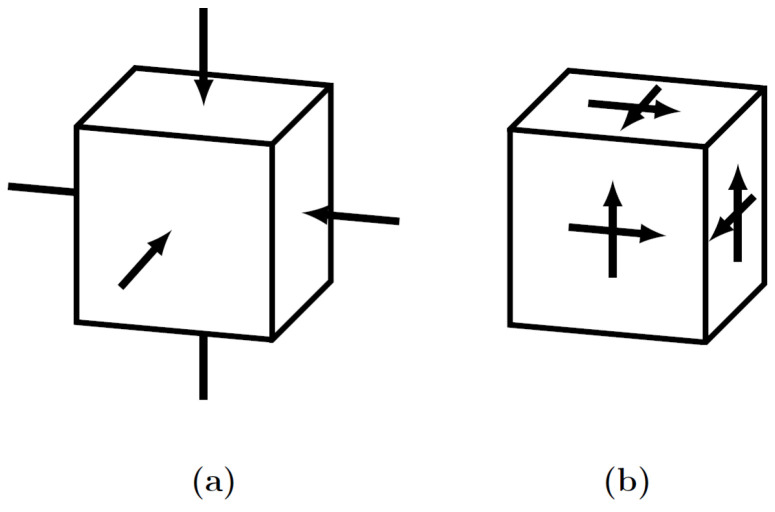
General definitions for PBC: (**a**) hydrostatic case; (**b**) shear case.

**Figure 4 polymers-13-01500-f004:**
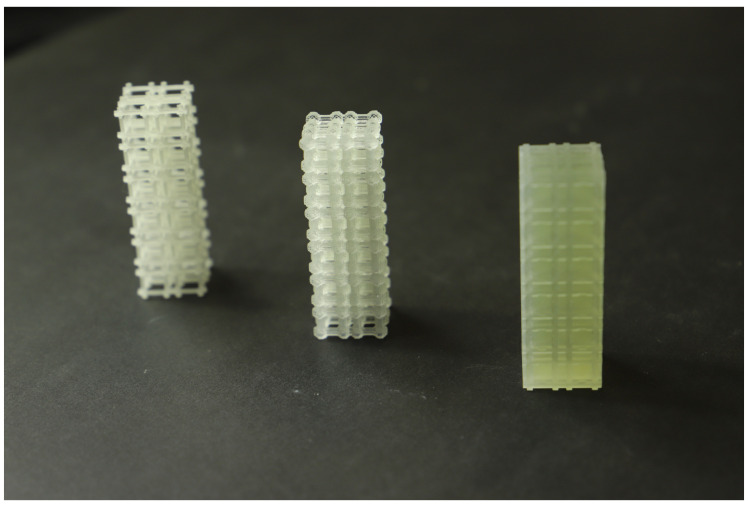
Test samples for maximum bulk: (left) 10$\%\;V_{f}$; (center) 30$\%\;V_{f}$ and (right) 60$\%\;V_{f}$.

**Figure 5 polymers-13-01500-f005:**
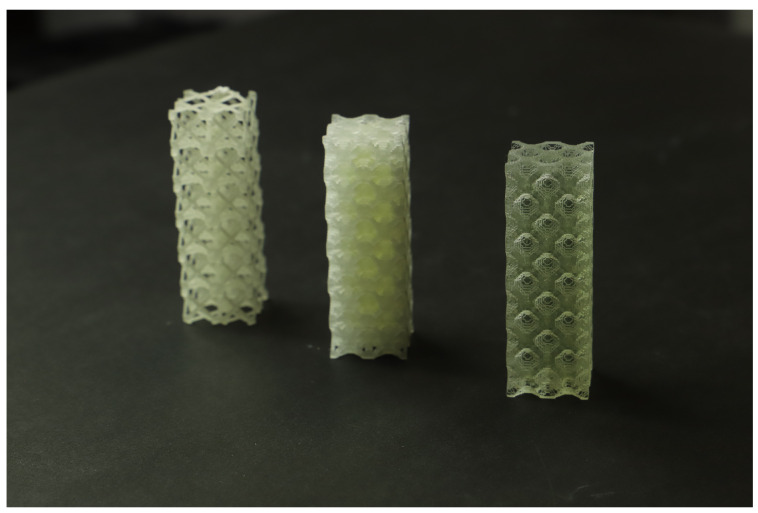
Test samples for maximum shear: (left) 10$\%\;V_{f}$; (center) 30$\%\;V_{f}$ and (right) 60$\%\;V_{f}$.

**Figure 6 polymers-13-01500-f006:**
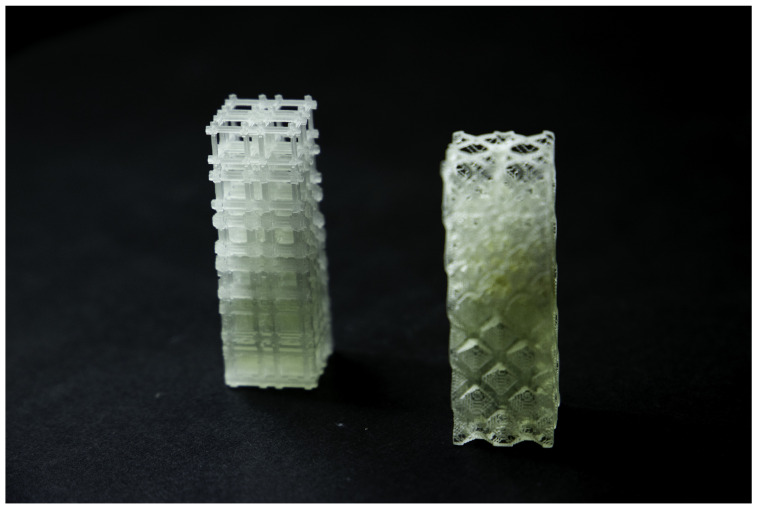
Test samples gradient case: (left) maximum bulk 10–60$\%\;V_{f}$; (right) maximum shear 10–60$\%\;V_{f}$.

**Figure 7 polymers-13-01500-f007:**
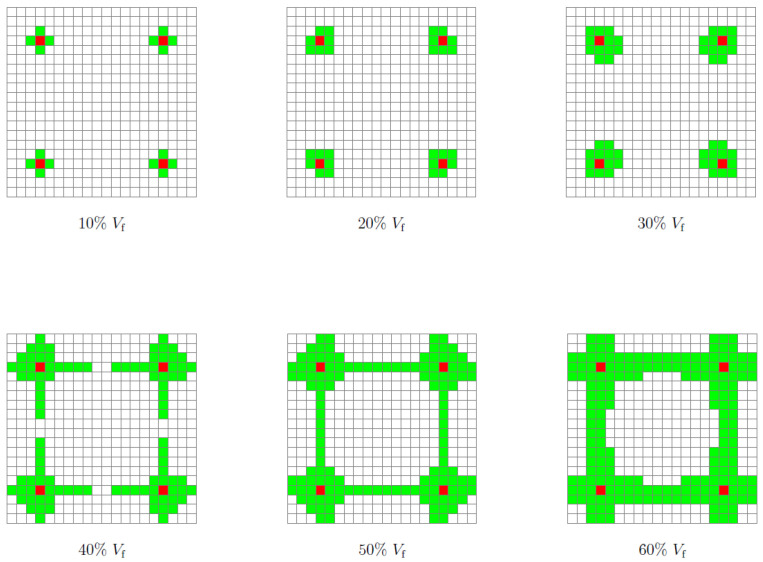
Kinematic connectors for maximum bulk 10, 20, 30, 40, 50, and 60$\%\;V_{f}$; green, boundary rigid elements; red, initial design domain.

**Figure 8 polymers-13-01500-f008:**
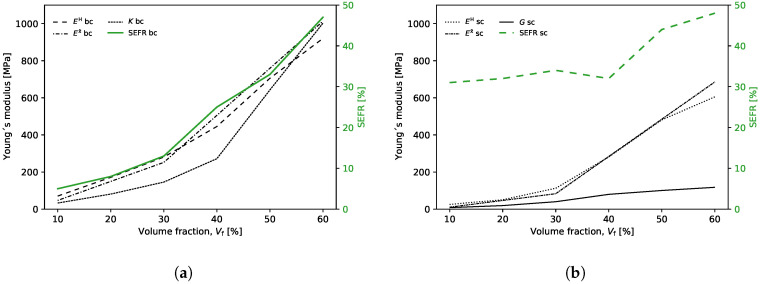
RUC’s homogenized Young’s modulus, bulk modulus, shear modulus and *SEFR* variation with volume fraction *V*_f_: (**a**) hydrostatic case; (**b**) shear case.

**Figure 9 polymers-13-01500-f009:**
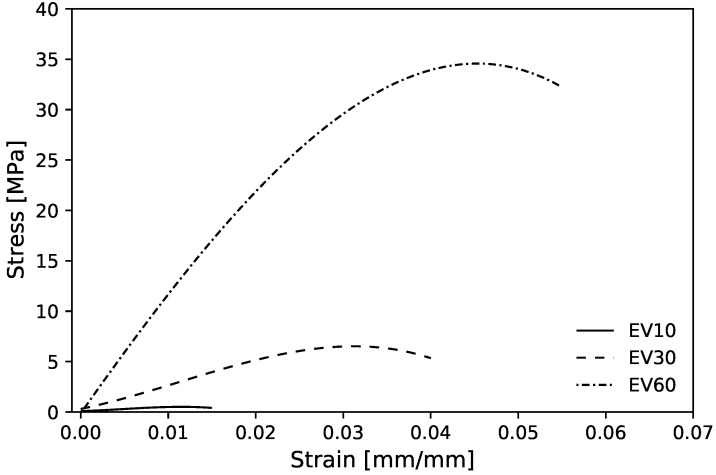
Experimental mean compressive stress–strain diagram for maximum bulk case with 10, 30 and 60$\%\;V_{f}$.

**Figure 10 polymers-13-01500-f010:**
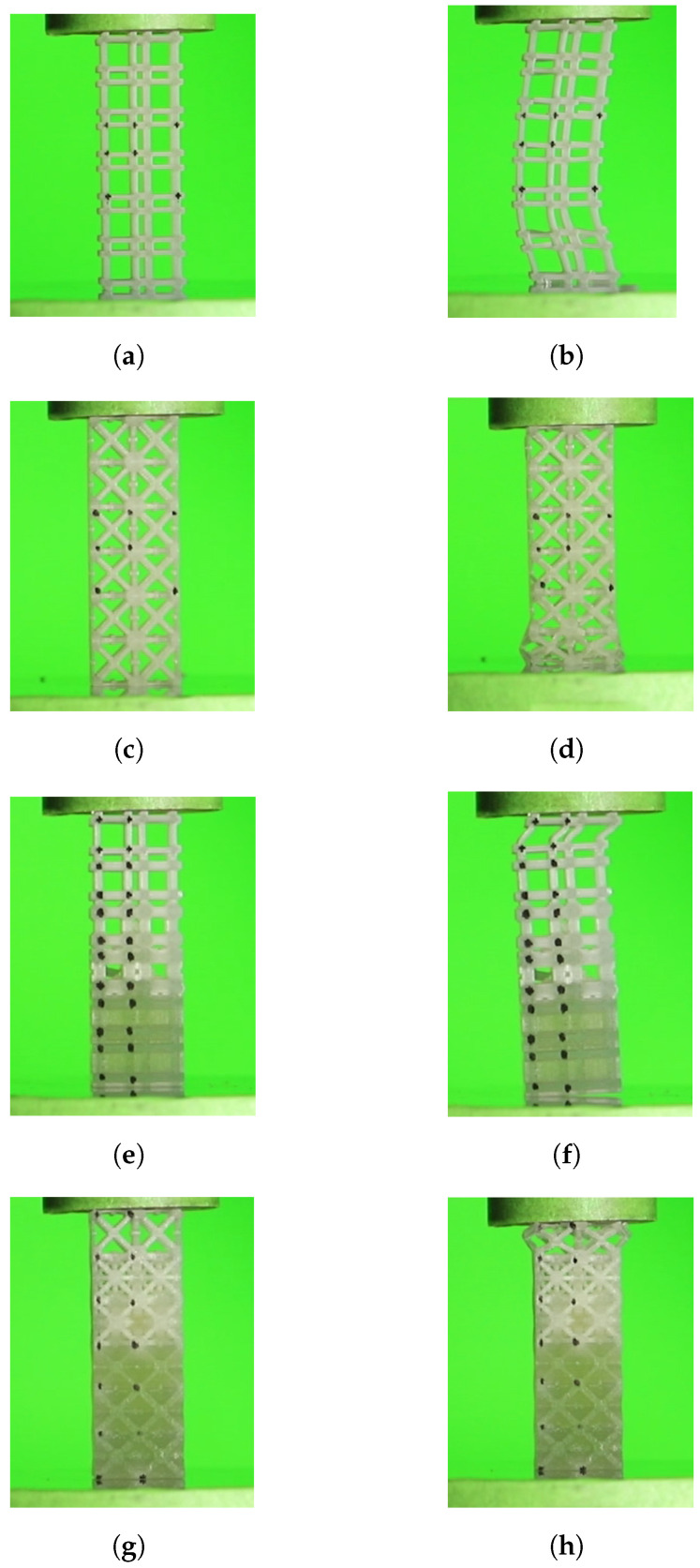
Compression test failure modes; (**a**) test sample for maximum bulk with 10% *V*_f_ not deformed; (**b**) test sample for maximum bulk with 10% *V*_f_ after a compression test, showing mechanical instability (buckling); (**c**) test sample for maximum shear with 10% *V*_f_ not deformed; (**d**) test sample for maximum shear with 10% *V*_f_ after a compression test, showing mechanical instability in lower test plate contact RUCs; (**e**) test sample gradient for maximum bulk with 10–60% *V*_f_ not deformed; (**f**) test sample gradient for maximum bulk with 10–60% *V*_f_ after a compression test, showing mechanical instability (buckling) in top test plate contact RUCs (10% *V*_f_); (**g**) test sample gradient for maximum shear with 10–60% *V*_f_ not deformed; (**h**) test sample gradient for maximum shear with 10–60% *V*_f_ after a compression test, showing mechanical instability in top test plate contact RUCs.

**Figure 11 polymers-13-01500-f011:**
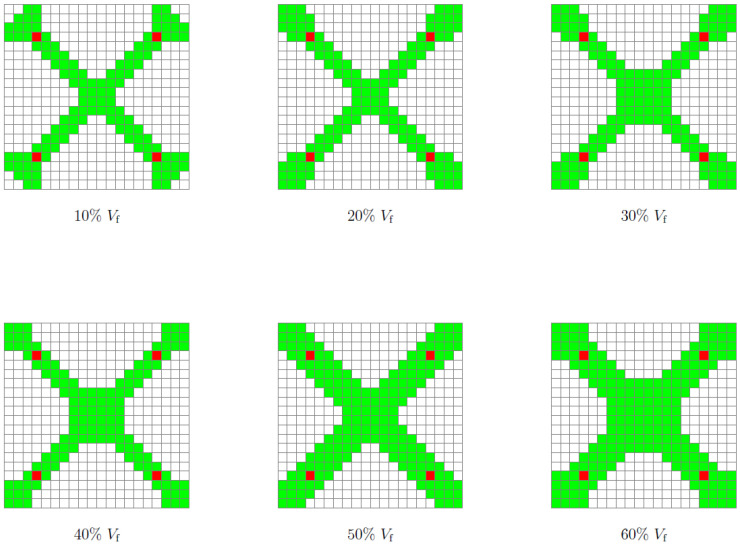
Kinematic connectors for maximum shear 10, 20, 30, 40, 50, and 60$\%\;V_{f}$; green, boundary rigid elements; red, initial design domain.

**Figure 12 polymers-13-01500-f012:**
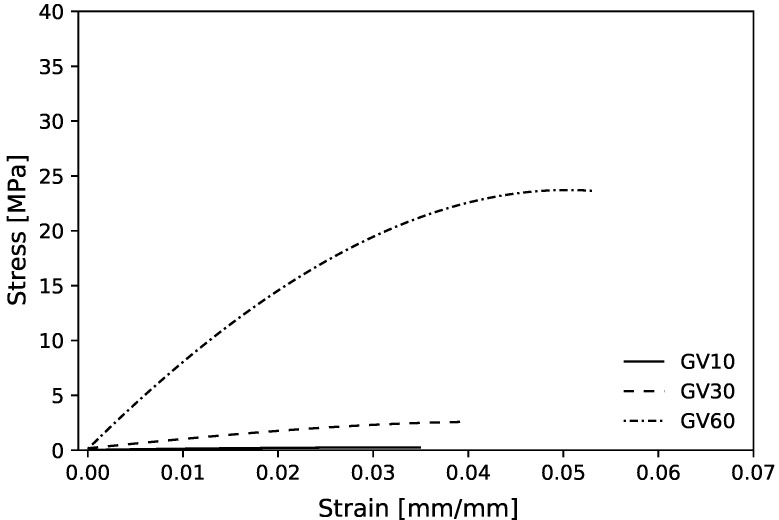
Experimental mean compressive stress–strain diagram for maximum shear case with 10, 30, and 60$\%\;V_{f}$.

**Figure 13 polymers-13-01500-f013:**
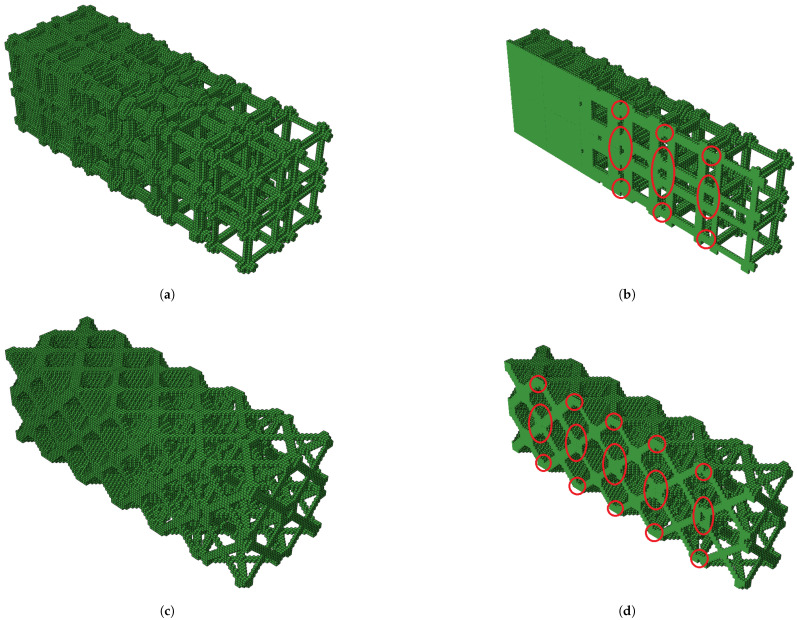
Graded hidrostactic case, 10–60% *V*_f_; (**a**) isometric view; (**b**) isometric view cut (red circles highlight interface between RUCs). Graded shear case 10–60% *V*_f_; (**c**) isometric view; (**d**) isometric view cut (red circles highlight interface between RUCs).

**Figure 14 polymers-13-01500-f014:**
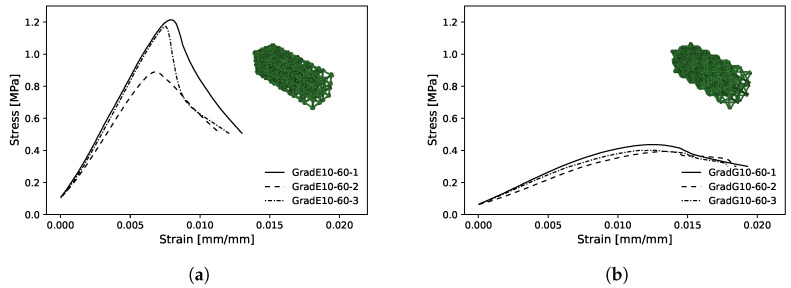
Experimental stress–strain diagram for: (**a**) maximum bulk case gradient 10–60% *V*_f_: (**b**) maximum shear case gradient 10–60% *V*_f_.

**Table 1 polymers-13-01500-t001:** SIMP parameters.

Volume Fraction, *V*_f_	10–60%
Penalty, *p*	4.0
Filter radius, $F_{r}$	2.0
Gray-scale filter, $G_{\mathrm{fr}}$	3.0
Material 1 (rigid)	*E* = 1850 MPa, ν = 0.49
Material 2 (support)	*E* = 0.3 MPa, ν = 0.49

**Table 2 polymers-13-01500-t002:** Experimental and homogenized RUC’s Young’s moduli.

Sample	$E^{E}$ [MPa]	$E^{\bar{x}}$ [MPa]	*SD* [MPa]	$E^{H}$ [MPa]	$E^{\mathrm{Se}}$ [MPa]	Err [%]
EV10-1	45.57					
EV10-2	52.49	47.09	3.94	70.53	—	−49.77
EV10-3	43.20					
EV30-1	256.61					
EV30-2	249.56	251.95	3.29	280.76	—	−11.43
EV30-3	251.95					
EV60-1	1015.55					
EV60-2	976.67	1012.69	28.32	919.99	—	9.15
EV60-3	1045.86					
GV10-1	11.48					
GV10-2	9.42	10.54	0.85	25.55	—	−142.41
GV10-3	10.72					
GV30-1	84.79					
GV30-2	80.39	83.11	1.94	112.56	—	−35.43
GV30-3	84.15					
GV60-1	680.61					
GV60-2	686.89	685.60	3.66	605.14	—	11.73
GV60-3	689.31					
GE1060-1	161.38					
GE1060-2	136.59	151.80	10.88	—	141.24	6.96
GE1060-3	157.43					
GG1060-1	41.78					
GG1060-2	33.30	37.70	3.47	—	53.13	−40.90
GG1060-3	38.02					

Note: EV10, hydrostatic sample 10% *V*_f_; EV30, hydrostatic sample 30% *V*_f_; EV60, hydrostatic sample 60% *V*_f_; GV10, shear sample 10% *V*_f_; GV30, shear sample 30% *V*_f_ and GV60, shear sample 60% *V*_f_; GE1060, hydrostatic graded sample 10–60% *V*_f_; GG1060, shear graded sample 10–60% *V*_f_; *E*^E^, experimental Young’s modulus; *E*^H^, homogenized Young’s modulus; *S*, standard deviation; *E*^Se^, theoretical total Young’s modulus of the graded structures; *E*^Se^, homogenized Young’s modulus; Err (%) = $\frac{E^{\mathrm{Measured}}-E^{\mathrm{Theoretical}}}{E^{\mathrm{Measured}}}$ × 100, relative error percentage.

**Table 3 polymers-13-01500-t003:** Design iterations and final properties of the RUC’s optimisation process.

Sample	No. Iter	*SEFR* [%]	*K* [MPa]	*G* [MPa]
EV10	59	5	32.93	—
EV20	40	8	81.17	—
EV30	47	13	145.77	—
EV40	43	25	271.93	—
EV50	50	33	642.38	—
EV60	46	47	1002.17	—
GV10	40	31	—	8.18
GV20	51	32	—	19.09
GV30	43	34	—	39.96
GV40	48	32	—	79.69
GV50	51	44	—	100.50
GV60	51	48	—	117.59

Note: EV10, hydrostatic sample 10% *V*_f_; EV20, hydrostatic sample 20% *V*_f_; EV30, hydrostatic sample 30% *V*_f_; EV40, hydrostatic sample 40% *V*_f_; EV50, hydrostatic sample 50% *V*_f_; EV60, hydrostatic sample 60% *V*_f_; GV10, shear sample 10% *V*_f_; GV20, shear sample 20% *V*_f_; GV30, shear sample 30% *V*_f_; GV40, shear sample 40% *V*_f_; GV50, shear sample 50% *V*_f_; and GV60, shear sample 60% *V*_f_; No. inter., number of iterations; *SEFR* = $\frac{number\;of\;solid\;elements\;per\;face}{total\;face\;elements}$, solid elements face ratio; *K*, bulk modulus; *G*, shear modulus.

## Data Availability

The data presented in this study are available on request from the corresponding author.

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
