# Peer review of "Design of Kinematic Connectors for Microstructured Materials Produced by Additive Manufacturing"

_polymers, 2021, doi:10.3390/polym13091500_

Round 1

Reviewer 1 Report

Dear Authors,

The presented manuscript consists a well-prepared work with a lot of experimental work and data analysis. However, the complexity of the studied item means that more elaboration on certain topics is needed. Please find below some comments about this work, as well as some references that it is suggested to be addressed at the mentioned points in the document:

  • What is the definition of the kinematic connector that is mentioned in the abstract as well as in the introduction? It is suggested to also include a schematic/figure of it in the introduction section along with a clear definition.
  • In the abstract, you mentioned that the proposed methodology is suitable for generating kinematic connections. However, it is not clear how the extracted results can be extrapolated to other designs as well as this work seems to rely on numerous experiments based on the optimization target (bulk and shear). In other scenario, the same methodology can be applicable?
  • In addition, according to the following reference, the user’s input and restrictions/constraints are critical steps during the Topology Optimization process. In your work you are mentioning the parameters that you are interested to optimize without mentioning the constraints. Please answer this point based on this work:
    • K. Lianos, H. Bikas, P. Stavropoulos, "A Shape Optimization Method for Part Design Derived from the Buildability Restrictions of the Directed Energy Deposition Additive Manufacturing Process", Designs, Volume 4, Issue 3, (2020)
  • Another significant part is the compromise between manufacturability and topology optimization based on the effort that is needed (energy, cost, time) to create the desired product. This topic can be considered during the initial steps of the topology optimization method. The following work describes the required steps of the manufacturability assessment prior to the mechanical tests that ask for final products and significant resources. Please consider this work when you answer:
    • K. Lianos, S. Koutsoukos, H. Bikas, P. Stavropoulos, "Manufacturability Assessment and Design for AM", Procedia CIRP, Volume 91, pg. 290-294,
  • In section 2.2, it is not mentioned if there was curing in the heated oven for the final resin parts. If yes, please provide the curing temperature and time as well as how significant is the curing for the specific resin in terms of yield strength.
  • Regarding the experimental work and the results what is the connection between the tested parts and the topology optimization procedure? The volume density of the part or the shape of the links between the layers and across each layer are results of the TO procedure? Please provide more information.
  • The process parameters of the AM process were constant during the creation of all parts?
  • Also, according to the following work that refers to mechanical testing of 3D printer parts, there are certain methodology and standards for this procedure. Have the authors considered any standard for this work? If no, please consider the following work to cover this question:
    • Vigneshwaran Shanmugam, Deepak Joel Johnson Rajendran, Karthik Babu, Sundarakannan Rajendran, Arumugaprabu Veerasimman, Uthayakumar Marimuthu, Sunpreet Singh, Oisik Das, Rasoul Esmaeely Neisiany, Mikael S. Hedenqvist, Filippo Berto, Seeram Ramakrishna, The mechanical testing and performance analysis of polymer-fibre composites prepared through the additive manufacturing, Polymer Testing,Volume 93, 2021, 106925, ISSN 0142-9418, https://doi.org/10.1016/j.polymertesting.2020.106925.

Thank you,

Reviewer 2 Report

This paper studies the kinematic connectors generated by imposing local density restrictions in the initial design domain between topologically optimized representative unit-cells. The authors studied different configurations for maximum bulk and shear moduli, varying the volume fractions and the Young’s modulus. Compression mechanical tests were performed to compare experimental and numerical Young’s modulus. The innovative proposed method proved to be suitable for generating kinematic connections for the design of graduated micro-structured materials.

Although this paper may be of real interest to industry experts, its reading is not easy and requires advanced skills in the field of materials and especially polymers. The authors considered many knowledges for normal and common to all readers, but, in my opinion, it is not so. A greater effort to make both the article and the experimental activities easier to understand could be done. The abstract is not so simple to understand, some numerical results would have much facilitated its understanding. The introduction is very long and dispersive. The authors should better clarify the novelties of the paper, the research design and the possible applications of the developed method. About the experimental section, it is necessary to better explain and motivate the choices regarding parameters, materials and methods of analysis. The conclusions need to be improved, because they are short and not very significative, not related to the numerical results of the paper. The possible applications of this study should be better emphasized, as well as the advantages that this method of analysis guarantees. Finally, the references section is complete, very extensive and well cared for, limiting self-citations to a minimum.

Below are all the main points to which the authors should reply:

  1. The abstract is very specific and difficult to understand in some parts. There are no numerical results, summarizing the paper. I recommend the authors to insert them, to facilitate the reader's understanding of the paper.
  2. At the end of the introduction, at lines 144-150, the targets of the papers, the novelties of the proposed method, the possible applications of this study should be better clarified. No indication on these points was provided in the introduction and they should be present.
  3. About lines 152-153, the authors should better explain the choice of the initial domain (cubic RUC), its size and its number of nodes, as well as the choice of using the 3D modeling SW Abaqus. The text should also be implemented in relation to this point.
  4. About table 1, at line 164, the choice of SIMP parameters should be much better clarified in the text. How were these parameters chosen? For which reason did the authors choose these values and not others?
  5. About line 193, the authors should motivate the choice of the polymeric material for the samples and supports. Why such materials? Is there a reason behind this choice?
  6. About line 194, the authors should explain the choice of 32 mm resolution. Is it a value set by the 3D machine? Or was it chosen by the authors? Was this parameter important for the purposes of the experimental analysis conducted?
  7. About line 203, the authors should motivate the choice of a test speed of 1 mm/min for mechanical compression tests. Why this value? Is it standard for testing?
  8. About lines 203-205, it is not clear how many replications of each test have been analyzed, nor how many are the total specimens. How do you get a total of 24? Which are the specimens? The authors on this point should make much more clarity within the text.
  9. About last column of table 2, there are very significant percentage errors on the test pieces EV-10, GV-10, GV-30 and GG-1060. Is such a situation possible? The reason for these values should be explained well in the text and if it were possible to reduce them in some way. They seem to me far from acceptable. The authors on this point should make much more clarity within the text.
  10. About table 3, explain why the specimens EV 20, 40, 50 and GV 20, 40, 50 are not also present in table 2.
  11. About table 3, the number of iterations and the SEFR value are also shown in the table. Why are these parameters important? From reading the text it is not at all clear and this lack could confuse the reader.
  12. At lines 239-240, the authors wrote: “These curves have characteristics of a semi-rigid polymer with an elastic region, however without showing a well-defined transition to plastic deformation region”. Why is this mechanical behavior present? Is there a reason behind this? The text should also be implemented in relation to this point.
  13. About lines 241-243, the authors referred to deviations of Table 2, but there are errors of greater magnitude than those reported. Why were they not taken into consideration?
  14. About figure 7a, there are no indications in the graph about the SD calculated in table 2. It should be inserted for a greater clarity and completeness of the figure.
  15. At lines 271-272, the authors wrote: “These curves have characteristics of a semi-rigid polymer with an elastic region, however without showing a well-defined transition to plastic deformation region”. Why is this mechanical behavior present? Also for this point, is there a reason behind this? The text should also be implemented in relation to this point.
  16. About figure 12a, you notice some very distant curves... is there a reason behind this behavior? Why does it happen? The text should also be implemented in relation to this point.
  17. About conclusions section, these are short, not very specific, not related to the numerical results obtained, not mentioned. I recommend implementing the section, explaining which are the advantages behind the developed method and which are the possible applications.
  18. Still in relation to conclusions, at lines 313-315, the authors wrote: "However, it can be discretized with different mesh sizes (i.e., coarse, or fine) and scale for a specific applications and others fabrication technologies". The authors should better clarify what they are referring to, i.e. which are the specific applications of these components and which alternative manufacturing technologies could be used to produce them.
